# Comparative Analysis of Bisexual and Parthenogenetic Populations in Haemaphysalis Longicornis

**DOI:** 10.3390/microorganisms12040823

**Published:** 2024-04-19

**Authors:** Chaoyue Zhao, Guonan Cai, Xing Zhang, Xinyu Liu, Pengfei Wang, Aihua Zheng

**Affiliations:** 1Shanghai Pudong Hospital, Fudan University Pudong Medical Center, School of Life Sciences, Fudan University, Shanghai 200437, China; chaoyuejaul@163.com (C.Z.); caiguonan480@gmail.com (G.C.); 18396701922@163.com (X.L.); 2State Key Laboratory of Integrated Management of Pest Insects and Rodents, Institute of Zoology, Chinese Academy of Sciences, Beijing 100101, China; zhangxing@ucas.ac.cn; 3Aulin College, Northeast Forestry University, Harbin 150040, China

**Keywords:** *Haemaphysalis longicornis*, parthenogenetic population, bisexual population, virus, transmission

## Abstract

*Haemaphysalis longicornis*, a three-host tick with a wide host range, is widely distributed in different countries and regions. It stands out among ticks due to its unique feature of having both parthenogenetic and bisexual populations. Despite their morphological resemblance, the characteristics of the parthenogenetic population have been overlooked. In this comprehensive study, we systematically compared the similarities and differences between these two populations. Our investigation revealed that the parthenogenetic *H. longicornis*, widely distributed in China, was found in ten provinces, surpassing the previously reported distribution. Notably, individuals from the parthenogenetic population exhibited a prolonged blood-feeding duration during the larval and nymph stages compared to their bisexual counterparts. Additionally, the life cycle of the parthenogenetic population was observed to be longer. A flow cytometry analysis indicated a DNA content ratio of approximately 2:3 between the bisexual and parthenogenetic populations. A phylogenetic analysis using whole mitochondrial genome sequences resulted in the separation of the phylogenetic tree into two distinct branches. A molecular analysis unveiled a consistent single T-base deletion at nucleotide 8497 in the parthenogenetic population compared to the bisexual population. Both populations displayed high viral infection capability and significant resistance to ivermectin. Intriguingly, despite these differences, the parthenogenetic population exhibited a similar life cycle to the bisexual population, retaining the ability to transmit pathogens such as Severe fever with thrombocytopenia syndrome virus (SFTSV) and Heartland Virus (HRTV). These findings contribute to a deeper understanding of the distinct characteristics and similarities between different populations of *H. longicornis*, laying the foundation for future research in this field.

## 1. Introduction

Ticks are obligate blood-sucking ectoparasites found globally, frequently parasitizing vertebrates. With over 960 species worldwide, ticks belong to various families, including *Ixodidae*, *Argasidae*, *Nuttalliellidae*, and *Deinocrotonidae* [1,2]. In China, at least 124 tick species have been recorded [1,3,4]. Tick-borne diseases (TBDs) caused by ticks, which are the second-largest pathogen vector in the world after mosquitoes, have resulted in billions of yuan in losses every year [5,6].

*Haemaphysalis longicornis*, native to eastern China, Japan, the Russian Far East, and South Korea, possesses the capability for rapid spread to new areas [7,8]. This species has been introduced as an exotic species in Australia, New Zealand, and various island countries in the western Pacific [9,10]. First identified in a sheep in New Jersey, USA, in 2017, *H. longicornis* has since been discovered in multiple eastern states of the United States [11,12]. As a vector for over 30 human or animal pathogens, including viruses, bacteria, and protozoa, it raises significant medical and veterinary concerns [8,13,14]. *H. longicornis* has been shown to be the vector of tick-borne encephalitis virus (TBEV) and (SFTSV) [11,15,16]. These tick-borne pathogenic microorganisms spread in a zoonotic way, often resulting in high mortality rates, which can cause significant damage to livestock economies [14,17,18].

Notably, *H. longicornis* is one of the few tick species with parthenogenetic and bisexual populations [15]. The parthenogenetic population originated in northern Japan and is now common in the Asia–Pacific region including China, Australia, New Zealand, New Caledonia, Fiji, New Hebrides, and Tonga [7,8,10]. While both parthenogenetic and bisexual populations are found in East Asia, only the parthenogenetic population is present in Oceania. In China, the parthenogenetic population has been identified in select areas, such as Shanghai city and Sichuan Province [19]. In the United States, the parthenogenetic population of *H. longicornis* was initially discovered in New Jersey in 2017, and by 2020, it had been found in 12 states, predominantly in the Eastern United States [12,20]. Distinguishing between the morphologies of these two reproductive groups is challenging, except for differences in Haller’s organ and the genital apron [19]. Recent studies utilizing scanning electron microscopy (SEM) have aided in describing the morphological characteristics of these reproductive populations [21]. Additionally, parthenogenetic *H. longicornis* exhibits larger body weight than bisexual *H. longicornis* at all developmental stages, and the former produces approximately twice as many offspring as the latter. A karyotype analysis revealed that the bisexual reproduction is diploid, while the parthenogenetic reproduction is triploid [10,22,23].

Significant efforts have been devoted to characterizing the taxonomic status, geographical distribution, morphological features, and chromosomal ploidy diversity of various *Haemaphysalis* species [10,15,19,21]. However, the parthenogenetic population of *H. longicornis* has exhibited widespread distribution in China and has demonstrated a higher dispersal rate compared to the bisexual population [24]. It is essential to comparatively analyze the difference between the two populations for basic research. Until now, there has been no study which has systematically investigated the biological parameters of both parthenogenetic and bisexual populations in parallel, which holds great significance. Therefore, in this study, the differences between the parthenogenetic and bisexual populations of *H. longicornis* were evaluated by comparing their distribution patterns, blood-meal times, and biological and molecular characteristics; meanwhile, our analysis of their life cycle, their ability to transmit pathogens, and their drug resistance showed that they are in close resemblance.

## 2. Materials and Methods

### 2.1. Ethics Statement

All animal studies were carried out in strict accordance with the recommendations in the Guide for the Care and Use of Laboratory Animals of the Ministry of Science and Technology of the People’s Republic of China. The protocols for animal studies were approved by the Committee on the Ethics of Animal Experiments of the Institute of Zoology, Chinese Academy of Sciences (Approval number: IOZ20180058).

### 2.2. Sample Collection

Ticks of all life stages were collected using drag cloth or removed from animals. From 2019 to 2022, we collected ticks (larvae, nymphs, and adults) in all regions of China except Xinjiang, Tibet, Qinghai, Ningxia, Inner Mongolia, Heilongjiang, Fujian, Guangxi, and Taiwan. The tick specimens were stored at −80 °C until used.

### 2.3. Identification of Tick Species and Phylogenetic Analysis

Ticks were identified based on morphological characteristics visualized through a light microscope with further molecular confirmation in the laboratory by sequencing the mitochondrial 16S ribosomal RNA (16S rRNA) gene. The primers were as follows: (16S-1) CTGCTCAATGATTTTTTAAATTGCTGTGG (Forward primer) and (16S-2) CGCTGTTATCCCTAGAGTATT (Reverse primer). A single leg was removed from each tick for the molecular analysis to confirm identification. A phylogenetic analysis was performed using whole mitochondrial genomes. Tick DNA was extracted using the MightyPrep reagent for DNA Kit (Takara, Beijing, China) according to the manufacturer’s instructions. The mitochondrial DNA was sequenced by next-generation sequencing (Tsingke Biotech, Beijing, China).

### 2.4. Polyploid Analysis of Ticks by FACS

Flow cytometry was used to test the ploidy of the tick chromosomes. Sysmex Partec CyFlow Space (Sysmex Partec, Görlitz, Germany) was used in this analysis and a Sysmex Partec CyStain DNA 1 Step kit was used. Briefly, a single live tick was placed in a plastic Petri dish and was disrupted by a blade in 400 μL of lysis buffer to fully extract the complete nuclei for 30 s. We filtered the liquid in the Petri dish with a Partec 30 μm filter and then added 1600 μL of dye and stained it for 10 s. Then, samples were tested on Sysmex Partec CyFlow Space.

### 2.5. Phylogenetic Tree and Genetic Distance

The phylogenetic tree was constructed using the maximum likelihood method with MEGA-X and the bootstrap value set at 1000.

### 2.6. Animal Experiment

*Atelerix albiventris* hedgehogs were purchased from Longchong Pet in Beijing. BALB/c mice and IFNAR^−/−^ C57/BL6 mice were purchased from Vitalriver (Beijing, China). Blood-feeding time was measured in BALB/c mice and *A. albiventris* hedgehogs. Bisexual and parthenogenetic larvae and nymph ticks were fed on BALB/c mice. Adult ticks were fed on *A. albiventris* hedgehogs. Engorged ticks were collected until the ticks detached from the host.

A virus infection assay was conducted in a bio-safety level 3 laboratory. Bisexual and parthenogenetic nymph ticks were fed on IFNAR^−/−^ C57/BL6 mice that were inoculated with 2 × 10^3^ fluorescence focus units (FFU) of SFTSV Wuhan strain until the ticks detached from the mice. SFTSV RNA copies were tested after molting into adult ticks.

For resistance tests, bisexual and parthenogenetic nymph ticks were fed on BALB/c mice that were fed with different concentrations of ivermectin-soaked rat chow. Survival rate, detachment speed, and engorged rate were observed.

### 2.7. HRTV and SFTSV RNA Extraction and Real-Time RT-PCR

Total RNA prepared from the homogenates of the ticks was extracted using TRIzol reagent (Thermo Fisher Scientific, Waltham, MA, USA) according to the manufacturer’s instructions. Samples were analyzed using a One-Step SYBR PrimerScript reverse transcription (RT)-PCR kit (TaKaRa, Beijing, China) on Applied Biosystems QuantStudio. Each sample was measured in triplicate. Conditions for the reaction were as follows: 42 °C for 5 min, 95 °C for 10 s, 40 cycles at 95 °C for 5 s, and 60 °C for 20 s.

### 2.8. Virus and Cells

HRTV, SFTSV Wuhan strain (GenBank accession numbers: S, KU361341.1; M, KU361342.1; L, KU361343.1), rabbit anti-SFTSV-NP polyclonal antibody was provided by the Wuhan Institute of Virology, Chinese Academy of Sciences. Vero cells (African green monkey kidney cells) were obtained from American Type Culture (ATCC) and maintained in Dulbecco’s modified Eagle’s medium (DMEM) supplemented with 8% fetal bovine serum (FBS), 1% L-glutamine, and 1% penicillin–streptomycin in a 37 °C incubator supplemented with 5% CO_2_. SFTSV was propagated at 37 °C in Vero cells at a multiplicity of infection of 0.1 and cultivated for 4 to 5 d. Cell culture supernatants containing viruses were collected and stored at −80 °C as the working stock for animal studies.

### 2.9. Virus Titration

A focus-forming assay was performed on Vero cells to titrate SFTSV. Vero cells were seeded in 96-well plates and reached ~90% confluence at the time of infection. The virus samples were diluted 10-fold in DMEM plus 2% FBS. After the removal of culture media, a diluted viral solution was added to the cells. Three hours later, the cells were washed once and incubated with DMEM plus 2% FBS and 20 mM NH4Cl at 37 °C. Two days post-infection, the cells were fixed with cold methanol and stained using a rabbit anti-SFTSV-NP polyclonal antibody at a 1:700 dilution and Alexa 488-labeled goat anti-rabbit IgG at a 1:700 dilution. Viral titers were examined under a fluorescent microscope and calculated by the Reed–Muench method.

A plaque-forming assay was performed on Vero cells to titrate HRTV titers. Vero cells were plated 24 h before infection in the 24-well plates at 2 × 10^4^ cells per well in triplicates. The virus samples were diluted 10-fold in DMEM with 2% FBS. After removal of medium, the cells were incubated with the 10-fold diluted viral solution at 37 °C. Three hours later, the cells were washed once and incubated with CMC-Hanks. The cells were incubated at 28 °C to allow plaques to form. The medium was aspirated, and the cells were stained with crystal violet. The titers were calculated by the Reed–Muench method.

## 3. Results

### 3.1. Subsection Geographical Distribution of H. longicornis

We conducted species identification on ticks collected from 19 provinces and 4 municipalities spanning the years 2019 to 2022, employing a combination of morphological and molecular biology approaches. *H. longicornis* was found in ticks from all provinces and municipalities except Hainan, Guangdong, and Guizhou Provinces (Figure 1A). The highest number of ticks was observed in Hainan Province, totaling 384 ticks, representing 17.3% of the overall count. Following closely were Beijing and Henan Provinces with 274 ticks (12.4%) and 201 ticks (9.08%), respectively. Conversely, Shanghai had the lowest tick count with only one, constituting 0.05% of the total ticks, trailed by Hunan Province and Chongqing city with two (0.09%) and eight (0.36%), respectively (Table 1). In the sampling area, a total of 1328 *H. longicornis* were collected, averaging 58 per province or municipality. Beijing had the highest tick count with 267, constituting 20.1% of the total ticks, followed by Henan Province and Shandong Province with 126 (9.49%) and 111 (8.36%), respectively. No *H. longicornis* were collected in Hainan, Guizhou, and Guangdong Provinces (Table 1). *H. longicornis* constituted 60.0% of the total tick count, with Jilin, Tianjin, Gansu, and Shanghai exclusively yielding *H. longicornis*. Yunnan Province had the lowest percentage of *H. longicornis* in the total tick count, followed by Chongqing city and Hunan Province with 13.6%, 37.5%, and 50.0%, respectively (Figure 1B).

### 3.2. Geographical Distribution of Bisexual and Parthenogenetic H. longicornis

We determined the reproductive patterns of 271 *H. longicornis* using flow cytometry and mitochondrial genome sequencing, as described in our previous study [24]. Among these, 186 individuals were identified as bisexual *H. longicornis*, constituting 69% of the total detected *H. longicornis*, while 85 were parthenogenetic *H. longicornis*, accounting for 31% (Figure 2A and Appendix A). The bisexual population was found in 15 provinces or municipalities, with exclusive identification in 5 of them. Beijing had the highest number of bisexual individuals, with 33 individuals detected, comprising 17.7% of the total diploids identified. This was followed by Liaoning Province and Hebei Province, with 24 (12.9%) and 26 (11.8%) individuals, respectively. No bisexual population was identified in Hunan, Jiangxi, Shanghai, and Yunnan Provinces or Chongqing city. A parthenogenetic population was detected in 15 provinces or municipalities, and 5 provinces or municipalities only had a parthenogenetic population. The parthenogenetic population detected was the same as the bisexual population. Henan Province had the highest number with 19, constituting 22.4% of the parthenogenetic individuals, followed by Shandong Province and Anhui Province with 13 (15.3%) and 12 (14.1%), respectively. No parthenogenetic population was identified in Hebei, Shaanxi, Shanxi and Jilin Provinces or Tianjin city. Both parthenogenesis and a bisexual population of *H. longicornis* were detected in 10 provinces or municipalities (Figure 2B).

### 3.3. Base Features and Blood-Sucking Time

*H. longicornis*, an obligate blood-feeding ixodid, follows a three-host life cycle and exhibits the capacity to parasitize a diverse range of native wildlife and livestock, including rodents, ungulates, lagomorphs, carnivores, and birds. Morphologically, the bisexual population and parthenogenetic populations of *H. longicornis* are similar, making them challenging to differentiate with the naked eye. Additionally, larvae and nymph ticks are difficult to distinguish based on size alone. Notably, during the adult stage, females of the bisexual population are relatively smaller than those of the parthenogenetic population. In laboratory settings, larvae and nymphs were fed on BALB/c mice, while adults were fed on *A. albiventris* hedgehogs (Figure 3A). During the larval stage, the mean blood-meal duration for parthenogenetic and bisexual populations was typically 5.3 and 5 days, respectively. In the nymph stage, the mean blood-meal duration for parthenogenetic and bisexual populations was equally 5.5 days. On the contrary, the mean blood-meal duration for the parthenogenetic and bisexual populations during the adult stage was 8 and 8.5 days, respectively (Figure 3A). Nevertheless, the weight of female bisexual individual after a blood meal increased by 50.4 times compared to pre-feeding weight, whereas the weight of male individuals remained nearly unchanged before and after blood-sucking. Moreover, the weight of parthenogenetic individuals after a blood meal increased by 157.8 times compared to the pre-feeding weight (Figure 3B).

### 3.4. Ploidy, Molecular, and Phylogenetic Analysis

We conducted a ploidy analysis on bisexual and parthenogenetic *H. longicornis* using FACS, evaluating the average fluorescence intensity as described in our previous study [24]. The ploidy test results revealed that the bisexual population exhibited a peak at position 51, while the parthenogenetic population showed a peak at position 75 (Figure 4A). To further investigate the genetic relationships, we selected 81 *H. longicornis* samples for the amplification of the complete mitochondrial genome, including 8 samples from overseas. The resulting phylogenetic tree demonstrated that the bisexual and parthenogenetic populations formed distinct lineages, suggesting an absence of gene exchange between these two populations (Figure 4B). Subsequently, the complete mitochondrial genome of *H. longicornis* was amplified and sequenced, revealing that the full genomes of bisexual and parthenogenetic *H. longicornis* were nearly identical at 14,694 and 14,693 bp, respectively. Notably, the parthenogenetic population had a deletion of a T base at position 8497, indicating a relatively conservative genetic structure (Figure 4C).

### 3.5. Susceptibility to HRTV and SFTS Virus

The foremost concern lies in discerning the variances in virus transmission capabilities between parthenogenetic and bisexual populations. In the case of HRTV, adult female ticks were infected with 1000 PFU through microinjection, and viral RNA copies were measured seven days post-infection. The mean titers for the bisexual and parthenogenetic populations were 4.7 log10 RNA copies/mg and 5.02 log10 RNA copies/mg, respectively, demonstrating no significant difference (Figure 5). These findings suggest comparable susceptibility to HRTV between the bisexual and parthenogenetic populations.

Moreover, we evaluated the ability of both populations to transmit SFTSV among hosts using a laboratory-adapted tick colony and an interferon α/β receptor knockout (IFNAR^−/−^) mouse model, as detailed in our prior study [24]. In brief, nymph ticks were infected by feeding on IFNAR^−/−^ C57/BL6 mice inoculated with 2 × 10^3^ FFU of SFTSV and collected after full engorgement. The average titer for both bisexual and parthenogenetic populations was 3 log10 RNA copies/μg, with no significant difference observed (Figure 6). In summary, these results indicate that the parthenogenetic population exhibits an equivalent ability to acquire and transmit SFTSV compared to the bisexual population.

### 3.6. Resistance to Ivermectin

Examining drug resistance in both parthenogenetic and bisexual populations is of critical concern. To assess resistance, BALB/c mice were fed with varying concentrations of ivermectin-soaked rat chow and dried hydatids. Subsequently, nymph ticks from both bisexual and parthenogenetic populations were allowed to feed on these mice, and the survival rate, detachment speed, and engorged rate of all ticks were observed. When the ivermectin dose was below 30 nL/g, all ticks on the mouse surface survived, with no significant difference in detachment speed observed between the bisexual and parthenogenetic populations. However, as the ivermectin dosage reached 30 nL/g, the detachment speed of ticks on the body surface slowed down, and none of the ticks became engorged. The detachment speed increased, but all ticks remained unfed and subsequently died as the ivermectin dosage reached 60 nL/g (Figure 7). These results indicate that there is no significant difference in drug resistance between the bisexual and parthenogenetic populations. Both populations exhibited a similar response to ivermectin, suggesting comparable levels of resistance in the face of varying drug concentrations.

## 4. Discussion

*H. longicornis* exhibits distinct reproductive modes, namely the bisexual and parthenogenetic populations. Initial studies showed that the parthenogenetic population only definitely occurred in Sichuan Province and Shanghai city [22,25,26]. In the present study, we conducted a nationwide collection of tick samples, finding *H. longicornis* in all surveyed locations except Hainan, Guizhou, and Guangdong Provinces. This comprehensive collection covered nearly the entire expanse of China. Surprisingly, we identified parthenogenetic populations in Beijing city, Shanghai city, and Henan, Shandong, Anhui, Liaoning, Hubei, Henan, Zhejiang, Jiangsu, Gansu, Anhui, Shandong, Yunnan, Sichuan, and Hunan Provinces. The broader distribution of the parthenogenetic population in China than previously reported suggests a wider prevalence. Furthermore, the distribution of the parthenogenetic population is closely linked to the endemic areas of SFTSV, as established in our previous study [24]. This reinforces the importance of not underestimating the significance of parthenogenetic populations.

In terms of biological characteristics, the blood-meal times for *H. longicornis* at each developmental stage align with parameters previously reported by Liu Jingze, Zhou Jinlin, and others [26,27]. Notably, the weight changes of larvae and nymph ticks before and after blood meals did not exhibit significant differences. However, some distinctions emerged in our investigations. The weight changes of fully engorged female bisexual individuals during the adult stage were noticeably smaller than those of parthenogenetic individuals, contrary to the characteristics reported by Zhou Jinlin [25]. Furthermore, male bisexual individuals were significantly lighter than their female counterparts. Previous research indicated that parthenogenesis results in double the offspring compared to bisexual reproduction [10,28]. Interestingly, the weight of parthenogenetic individuals was considerably higher than that of bisexual individuals, suggesting that the number of offspring produced by parthenogenetic individuals might surpass double that of bisexual individuals. This observation supports the idea that parthenogenetic populations may possess evolutionary advantages over bisexual populations.

The phylogenetic tree constructed based on partial mitochondrial genome sequences revealed that the two reproductive populations of *H. longicornis* were closely related, falling below the criteria for subspecies identification [4,29,30]. However, our analysis established a phylogenetic tree using the complete mitochondrial genome sequences of parthenogenetic and bisexual *H. longicornis*. The results indicated that parthenogenetic and bisexual populations formed two distinct lineages, suggesting that parthenogenetic populations may not have gene communication with bisexual populations. Additionally, the conservative deletion of a single T base at nucleotide 8497 in parthenogenetic populations could serve as a reliable marker for determining the reproductive mode of *H. longicornis*. This method proves to be more accurate compared to morphological analysis and more cost effective than ploidy analysis.

SFTSV is an emerging tick-borne phlebovirus first described in China in 2009, transmitted by *H. longicornis*. Between 2011 and 2018, cases of SFTS were reported in 21 of the 34 provinces in China, as well as in other Asian countries such as Korea, Japan, Vietnam, and Pakistan [24]. HRTV was identified and isolated from two patients in the United States in 2012. *H. longicornis* has been found in a total of 12 states in the Eastern and Central United States, and HRTV shares genetic similarities with SFTSV [31,32]. After injecting SFTSV ticks with HRTV, high levels of viral RNA were detected in *H. longicornis*, indicating that HRTV could be obtained by the ticks. Given that *H. longicornis* has been identified in most states in the Eastern and Central United States, there is a potential risk of the tick spreading along with HRTV.

*H. longicornis* is related to the spread of blood parasites (*Theileria Oriental*) throughout the Asian–Pacific region. It is reported that *H. longicornis* was introduced to Australia from northern Japan in the 19th century, and then spread to New Zealand, New Caledonia, and Fiji [33]. In recent years, *T. orientalis* has spread rapidly in the Asia-Pacific region and North America, indicating a high risk in the rapid spread of disease vectors. *H. longicornis* has also been shown to be susceptible to Rickettsia rickettsii, the agent of Rocky Mountain spotted fever, under laboratory conditions [34]. Given the ongoing globalization of the economy, increased trade, the movement of goods and people, escalating global climate change, and rapid urbanization, the spread of vector organisms and their transmitted pathogens has become a global phenomenon. The parthenogenetic population of *H. longicornis* and the pathogens it transmits may be undergoing a process of global dissemination. Future research should aim at enhancing our understanding of this phenomenon, investigating aspects such as the mechanism of tick parthenogenesis, the evolution of parthenogenetic tick populations, and the intricate interactions among the vector, pathogen, and host. This knowledge is crucial for devising effective strategies for mitigating the impact of emerging infectious diseases associated with ticks.

## 5. Conclusions

This study systematically compared the similarities and differences between parthenogenetic and bisexual populations of *H. longicornis*. The discovery of the broader distribution of *H. longicornis* throughout China and its relationship with SFTSV highlighted its potential in disease transmission. The biological characteristics and phylogenetic analysis revealed significant differences between the parthenogenetic and bisexual populations, suggesting a lack of gene flow and possible evolutionary advantages of parthenogenetic reproduction. The association of *H. longicornis* with blood parasites such as *Theileria oriental* and *Rickettsia rickettsii* highlighted its role as a vector in disease transmission. It is critical to understand the mechanisms behind tick parthenogenesis, the evolutionary dynamics of tick populations, and the complex interactions between vectors, pathogens, and hosts.

## Figures and Tables

**Figure 1 microorganisms-12-00823-f001:**
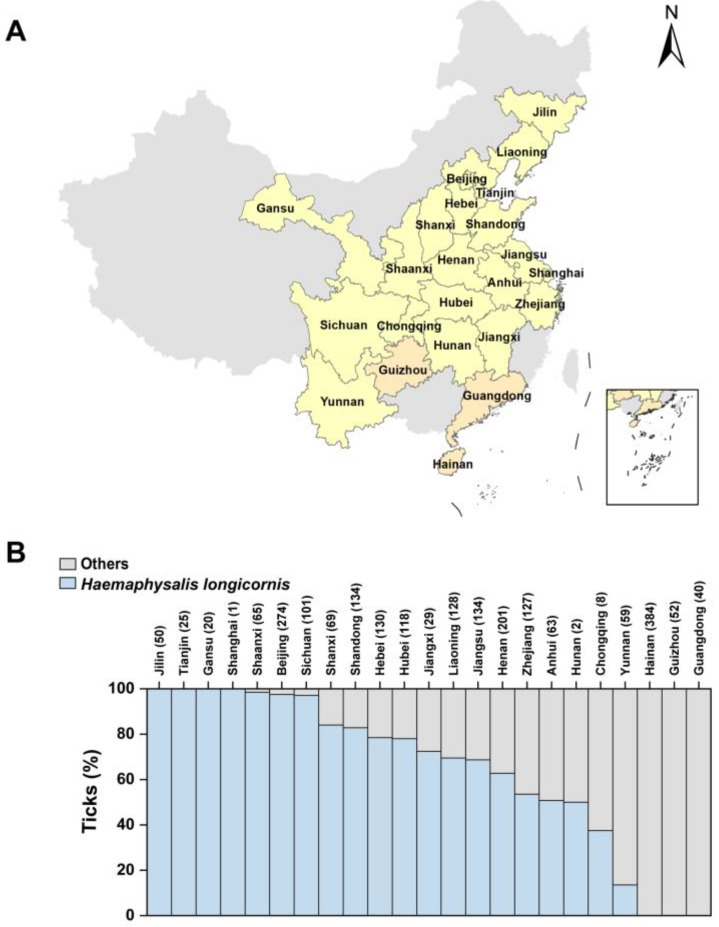
Geographical distribution of ticks collected in China. (**A**) Tick collection and distribution. The colored areas are the provinces where ticks were collected. Bright yellow areas are the provinces where *Haemaphysalis longicornis* were collected, while dark yellow ones are the provinces where they were not. (**B**) Percentage of *H. longicornis* collected in total ticks in each province. ‘Others’ represent ticks that are not *H. longicornis*. The numbers in parentheses after each province represent the total number of ticks collected in that province.

**Figure 2 microorganisms-12-00823-f002:**
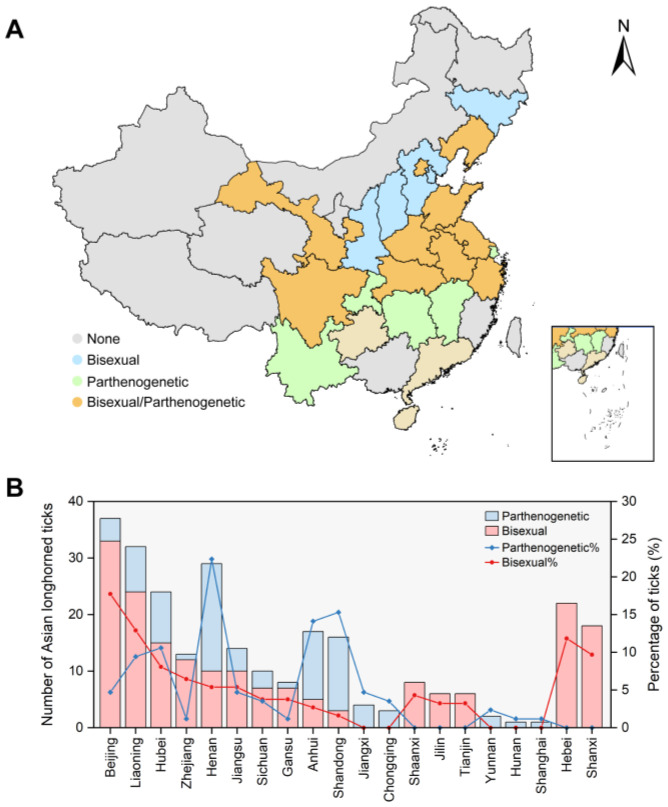
Geographical distribution of bisexual and parthenogenetic *Haemaphysalis longicornis* collected in China. (**A**) Distribution of bisexual and parthenogenetic *H. longicornis* in China. Green areas indicate parthenogenetic *H. longicornis*, blue areas indicate bisexual *H. longicornis*, brown areas indicate both bisexual and parthenogenetic *H. longicornis*, and grey areas indicate that there were no *H. longicornis* collected. (**B**) Distribution of bisexual and parthenogenetic *H. longicornis* in different provinces. The number and percentage of *H. longicornis* with different reproductive styles. Red represents bisexual *H. longicornis* (diploid) and blue represents parthenogenetic *H. longicornis* (triploid).

**Figure 3 microorganisms-12-00823-f003:**
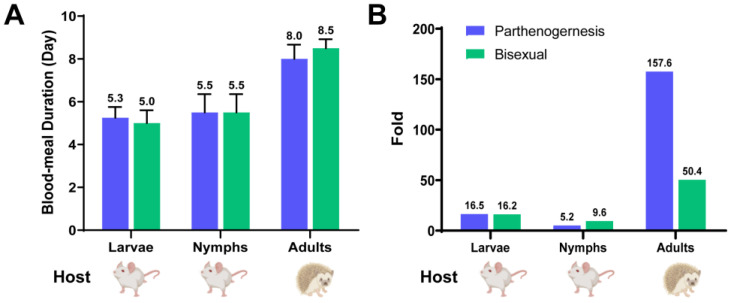
Differences in morphology and blood-meal time between bisexual and parthenogenetic *Haemaphysalis longicornis*. (**A**) Comparison of blood-meal time between bisexual and parthenogenetic *H. longicornis*. Green indicates the blood-meal time of bisexual *H. longicornis* and blue indicates the blood-meal time of parthenogenetic *H. longicornis*. The arrow indicates feeding with it. (**B**) Changes in body weight of bisexual and parthenogenetic *H. longicornis*. Green represents bisexual *H. longicornis*, blue represents parthenogenetic *H. longicornis*, and gray represents bisexual adult male *H. longicornis*.

**Figure 4 microorganisms-12-00823-f004:**
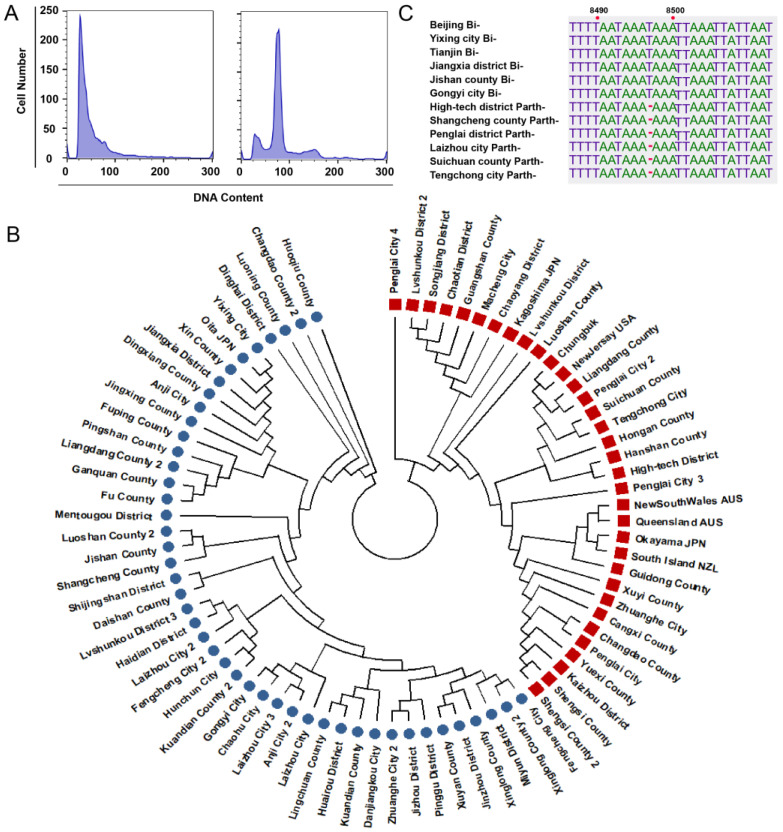
Differences in ploidy and genome between bisexual and parthenogenetic *Haemaphysalis longicornis*. (**A**) Flow cytometry histograms of bisexual and parthenogenetic *H. longicornis*. Ploidy of *H. longicornis* was tested by using flow cytometry and measuring the fluorescence intensity of cell nuclei stained with 4′,6-diamidino-2-phenylindole. DNA content per cell frequency distribution of bisexual and parthenogenetic *H. longicornis*. The relative DNA content in the G1 phase (the first peak) for bisexual population was 51 and was 75 for the parthenogenetic population. (**B**) Phylogenetic analysis of bisexual and parthenogenetic *H. longicornis*. Phylogenetic analysis of bisexual (blue) and parthenogenetic (red) *H. longicornis* in the Asia–Pacific region. Maximum likelihood trees were established with mitochondrial genomes of ticks collected in the Asia-Pacific region. Numbers indicate multiple *H. longicornis* from the same county. (**C**) Mitochondrial genome alignment between bisexual and parthenogenetic *H. longicornis*. The upper six sequences are from bisexual *H. longicornis* and the lower six sequences are from parthenogenetic *H. longicornis*.

**Figure 5 microorganisms-12-00823-f005:**
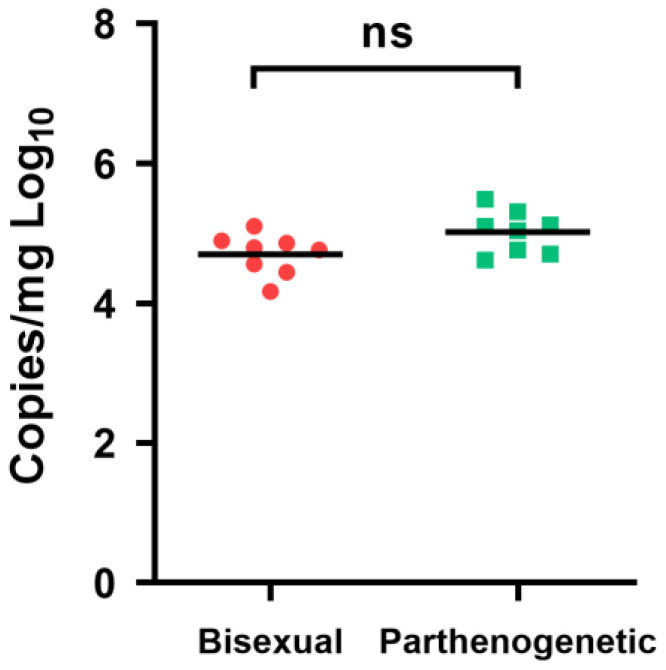
Susceptibility of bisexual and parthenogenetic *Haemaphysalis longicornis* to HRTV virus. Groups of bisexual and parthenogenetic *H. longicornis* adult ticks were infected with 1000 PFU of HRTV by microinjection. Seven days post-infection, total RNA samples from 8 ticks were extracted, and the viral RNA levels were detected by real-time PCR (n = 3). Each dot represents one tick. ns, not significant.

**Figure 6 microorganisms-12-00823-f006:**
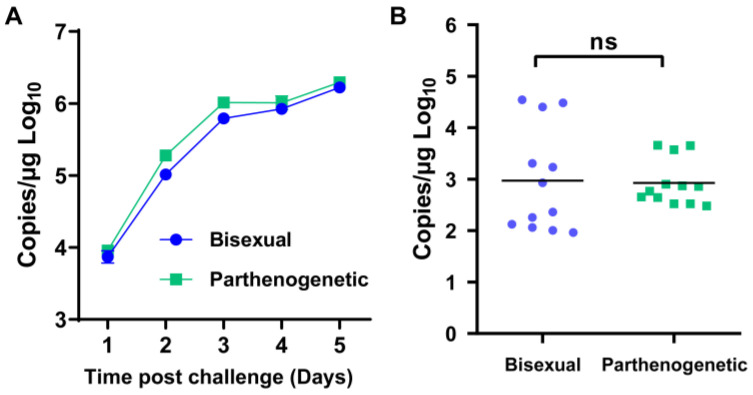
Susceptibility of bisexual and parthenogenetic *Haemaphysalis longicornis* to SFTS virus. Groups of bisexual and parthenogenetic *H. longicornis* nymph ticks were fed separately on one IFNAR^−/−^ C57/BL6 mouse that was intraperitoneally inoculated with 2 × 10^3^ FFU of SFTSV. (**A**) Viremia of IFNAR^−/−^ C57/BL6 mice was monitored by real-time PCR during tick feeding. (**B**) SFTSV RNA of the ticks was tested by real-time PCR after they molted into adults. Each dot or square indicates one tick. Black horizontal bars indicate means. ns, not significant.

**Figure 7 microorganisms-12-00823-f007:**
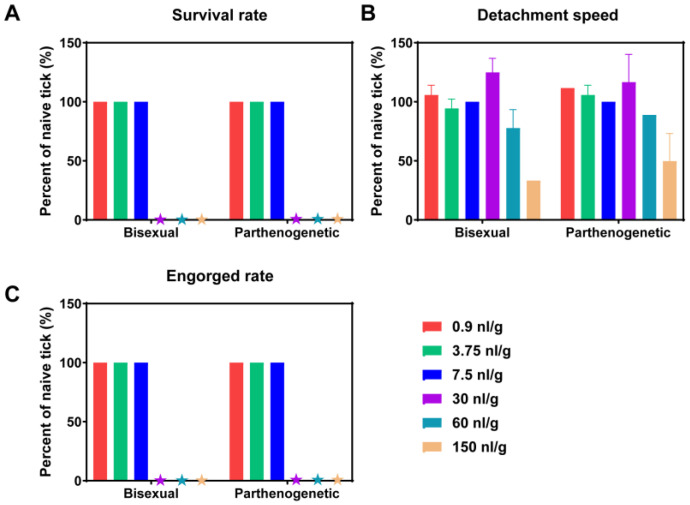
Resistance of bisexual and parthenogenetic *Haemaphysalis longicornis* to ivermectin on the small mammals. BALB/c mice were fed with different concentrations of ivermectin-soaked rat chow and dried hydatids, and the items of *H. longicornis* were observed. (**A**) Survival rate, (**B**) detachment speed, and (**C**) engorged rate relative to untreated ticks.

**Table 1 microorganisms-12-00823-t001:** Percentage of ticks collected in total ticks in each province and percentage of *H. longicornis* collected in total ticks in each province.

Province/Region	Number of Ticks	Number of *H. longicornis*	Percentage of Total Ticks	Percentage of Total *H. longicornis*	*H. longicornis* Percentage of Ticks
Jilin	50	50	2.26%	3.77%	100.00%
Tianjin	25	25	1.13%	1.88%	100.00%
Gansu	20	20	0.90%	1.51%	100.00%
Shanghai	1	1	0.05%	0.08%	100.00%
Shaanxi	65	64	2.94%	4.82%	98.46%
Beijing	274	267	12.38%	20.11%	97.45%
Sichuan	101	98	4.56%	7.38%	97.03%
Shanxi	69	58	3.12%	4.37%	84.06%
Shandong	134	111	6.05%	8.36%	82.84%
Hebei	130	102	5.87%	7.68%	78.46%
Hubei	118	92	5.33%	6.93%	77.97%
Jiangxi	29	21	1.31%	1.58%	72.41%
Liaoning	128	89	5.78%	6.70%	69.53%
Jiangsu	134	92	6.05%	6.93%	68.66%
Henan	201	126	9.08%	9.49%	62.69%
Zhejiang	127	68	5.74%	5.12%	53.54%
Anhui	63	32	2.85%	2.41%	50.79%
Hunan	2	1	0.09%	0.08%	50.00%
Chongqing	8	3	0.36%	0.23%	37.50%
Yunnan	59	8	2.66%	0.60%	13.56%
Hainan	384	0	17.34%	0.00%	0.00%
Guizhou	52	0	2.35%	0.00%	0.00%
Guangdong	40	0	1.81%	0.00%	0.00%
Total	2214	1328	100.00%	100.00%	59.98%

## Data Availability

Data are contained within the article and Appendix A.

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
