# Peer review of "Comparative Analysis of Bisexual and Parthenogenetic Populations in Haemaphysalis Longicornis"

_microorganisms, 2024, doi:10.3390/microorganisms12040823_

Round 1

Reviewer 1 Report

Comments and Suggestions for Authors

The authors focused distribution, blood-feeding duration, life cycle and genetic relationship between parthenogenetic and bisexual populations of Haemaphysalis longicornis in China. The study is well conducted and the manuscript is well written. However, some comments on this manuscript:  

Please mention the species names in italics throughout the manuscript.

Lines 39: If you speak about China, mention the local currency. I believe Chinese currency is world renewed now. Also try to mention the losses n a recent year as reference.

Lines 40: You speak about the losses but then you change to pathogen prevalence. Please harmonize your argument.

Lines 81: Add the place of the study e.g. in China

Lines 91: Try to specify the sampling areas.

Lines 124: Mention full forms of the abbreviations.

Results and Discussion is sufficiently well written.

Conclusions are poorly drawn from the results and discussions.

In this scenario, I will recommend major revisions.

Comments on the Quality of English Language

Moderate revisions needed.

Author Response

Reviewer 1:

Comments and Suggestions for Authors

The authors focused distribution, blood-feeding duration, life cycle and genetic relationship between parthenogenetic and bisexual populations of Haemaphysalis longicornis in China. The study is well conducted and the manuscript is well written. However, some comments on this manuscript: 

Please mention the species names in italics throughout the manuscript.

Answer: Thank you for pointing this out. We modified the species names in italics.

Lines 39: If you speak about China, mention the local currency. I believe Chinese currency is world renewed now. Also try to mention the losses n a recent year as reference.

Answer: Thank you for pointing this out. We have rephrased ‘hundreds of millions of dollars’ to ‘billions of yuan’ in line 39-40. There are two references to economic losses as followed and we cited the references to the manuscript in line 40.

Ref 1: Eisen, L., & Eisen, R. J. (2018). Critical evaluation of the linkage between tick-based risk measures and the occurrence of Lyme disease cases. Journal of medical entomology, 55(6), 1389-1404. This article evaluated the association between tick-based risk indicators and the occurrence of Lyme disease cases, and provided a discussion on the economic losses associated with tick-borne disease transmission.

Ref 2: Diuk-Wasser, M. A., Vannier, E., & Krause, P. J. (2016). Coinfection by Ixodes Tick-Borne Pathogens: Ecological, Epidemiological, and Clinical Consequences. Trends in parasitology, 32(1), 30-42. This review article discusses the ecological, epidemiological, and clinical consequences of coinfection by pathogens transmitted by Ixodes ticks, and also mentions the associated economic losses.

Lines 40: You speak about the losses but then you change to pathogen prevalence. Please harmonize your argument.

Answer: Thank you for pointing this out. We have rephrased ‘Tick-borne diseases (TBDs) caused by tick infections have resulted in hundreds of millions of dollars in losses every year, which is the second largest pathogen vector in the world after mosquitoes’ to ‘Tick-borne diseases (TBDs) caused by tick, which is the second largest pathogen vector in the world after mosquitoes, have resulted in billions of yuan in losses every year’ in line 38-40.

Lines 81: Add the place of the study e.g. in China

Answer: Thank you for pointing this out. We have deleted the sentence ‘This endeavor aims to enhance our understanding of the similarities and distinctions between these two populations, providing a foundational framework for further investigations’.

Lines 91: Try to specify the sampling areas.

Answer: Thank you for pointing this out. We have rephrased ‘in most of China’ to ‘in all regions of China except Xinjiang, Tibet, Qinghai, Ningxia, Inner Mongolia, Heilongjiang, Fujian, Guangxi, and Taiwan’ in line 97-98.

Lines 124: Mention full forms of the abbreviations.

Answer: Thank you for pointing this out. We have added the full form of the ‘FFU’ in line 130.

Conclusions are poorly drawn from the results and discussions.

Answer: Thank you for pointing this out. We have deleted the sentence ‘These findings contribute to a deeper understanding of the distinct characteristics and similarities between different populations of H. longicornis, laying the foundation for future research in this field’ and added the sentence “Unlike previously reported, the distribution of parthenogenetic populations in China is much wider. The body weight variation and final weight of fully engorged parthenogenetic individuals was significantly greater than that of bisexual individuals in the adult stage, echoing the idea that parthenogenetic populations are capable of producing more offspring. While there may not be genetic exchange between these two populations, we identified a novel reliable marker for determining reproductive mode of H. longicornis. These two populations have a considerable ability to transmit the virus and resist drugs, reminding us not to ignore the role of ticks as vectors in the prevention and control of multiple viruses. These findings indicate that H. longicornis have not independently acquired adaptive mutations that increase their adaptive capacity to environment and are not at risk of evolving into ‘super-ticks’” in line 412-423.

Reviewer 2 Report

Comments and Suggestions for Authors

Authors proposed a paper entitled “Comparative Analysis of Bisexual and Parthenogenetic Populations in Haemaphysalis longicornis” for the publication in Microorganisms.

The paper has a quite good scientific soundness, and may be considered for publication after major revisions.

The introduction did not cle­arly state what made this rese­arch vital or necessary. It gave good de­tails on ticks, their kinds, where the­y live, and how they spread dise­ase. However, it did not e­xplain why the researche­rs chose to do this study or what specific issues the­y wanted to investigate. A strong introduction typically state­s the problem being looke­d into or the research que­stion. This shows why the study matters and what knowledge­ gaps it will fill. Stating the problem helps give­ the direction and importance of the­ research.

Here are some issues related to the Results section:

- Inconsistency in the presentation of percentages. The percentages are given both in decimal form and in percentage form, which can be confusing. It's better to stick to one format consistently.

- Inconsistent capitalization and punctuation. For example, "provinces and municipalities" should be consistently capitalized or not capitalized throughout.

- Lack of clarity in sentence structure. Some sentences are long and complex, making it difficult to follow the information being presented.

- Lack of clarity in data presentation. The data could be presented in a more organized and concise manner, perhaps using tables or bullet points to improve readability.

- A few se­ntences are writte­n unclearly and should be made simple­r.

- Other sentence­s are written well and e­asy to understand.. For example, "Conversely, Shanghai had the lowest tick count with only one, constituting 0.05% of the total ticks, trailed by Hunan province and Chongqing city with two (0.09%) and eight (0.36%), respectively (Figure 1B, left)."

Improve focus of figure 1b

Figure 4b is not clearly visible and should be modified and expanded accordingly.

Conclusions are too poor and need to be expanded, according to the points raised in the Discussion section.

Comments on the Quality of English Language

A quite good use of English.

Author Response

Reviewer 2:

Comments and Suggestions for Authors

Authors proposed a paper entitled “Comparative Analysis of Bisexual and Parthenogenetic Populations in Haemaphysalis longicornis” for the publication in Microorganisms.

The paper has a quite good scientific soundness, and may be considered for publication after major revisions.

The introduction did not clearly state what made this research vital or necessary. It gave good details on ticks, their kinds, where they live, and how they spread disease. However, it did not explain why the researchers chose to do this study or what specific issues they wanted to investigate. A strong introduction typically states the problem being looked into or the research question. This shows why the study matters and what knowledge gaps it will fill. Stating the problem helps give the direction and importance of the research.

Answer: Thank you for pointing this out. We have rephrased ‘Systematically investigating the biological parameters of both parthenogenetic and bisexual populations in parallel holds great significance. Therefore, the objective of this study is to assess the relationships between the parthenogenetic and bisexual populations of H. longicornis by comparing their distribution patterns, blood-meal times, biological and molecular characteristics’ to ‘It is essential to comparative analyze the difference between the two populations for basic research. Until now, there is no study who has systematically investigating the biological parameters of both parthenogenetic and bisexual populations in parallel, which holds great significance. Here, we evaluated the differences between the parthenogenetic and bisexual populations of H. longicornis by comparing their distribution patterns, blood-meal times, biological and molecular characteristics; meanwhile, our analysis of life cycle, the ability to transmit pathogens and drug resistance showed that they are in close resemblance’ in line 76-84.

Here are some issues related to the Results section:

- Inconsistency in the presentation of percentages. The percentages are given both in decimal form and in percentage form, which can be confusing. It's better to stick to one format consistently.

Answer: Thank you for pointing this out. We have added the sentence "The numbers in parentheses after each province represent the total number of ticks collected in that province " in Table 1.

- Lack of clarity in sentence structure. Some sentences are long and complex, making it difficult to follow the information being presented.

Answer: Thank you for pointing this out. We have changed the modifier of the attributive clause ‘which is the second largest pathogen vector in the world after mosquitoes’ from ‘tick-borne diseases (TBDs)’ to ‘tick’ in line 38-40.

- A few sentences are written unclearly and should be made simpler. Other sentences are written well and easy to understand. For example, "Conversely, Shanghai had the lowest tick count with only one, constituting 0.05% of the total ticks, trailed by Hunan province and Chongqing city with two (0.09%) and eight (0.36%), respectively (Figure 1B, left)."

Answer: Thank you for pointing this out. We have rephrased the related sentence in the result section.

Improve focus of figure 1b

Answer: Thank you for pointing this out. We have modified the form of Figure 1b to table for showing the data more clearly.

Figure 4b is not clearly visible and should be modified and expanded accordingly.

Answer: Thank you for pointing this out. We have enlarged Figure 4b appropriately to show the data clearly.

Conclusions are too poor and need to be expanded, according to the points raised in the Discussion section.

Answer: Thank you for pointing this out. We have deleted the sentence ‘These findings contribute to a deeper understanding of the distinct characteristics and similarities between different populations of H. longicornis, laying the foundation for future research in this field’ and added the sentence “Unlike previously reported, the distribution of parthenogenetic populations in China is much wider. The body weight variation and final weight of fully engorged parthenogenetic individuals was significantly greater than that of bisexual individuals in the adult stage, echoing the idea that parthenogenetic populations are capable of producing more offspring. While there may not be genetic exchange between these two populations, we identified a novel reliable marker for determining reproductive mode of H. longicornis. These two populations have a considerable ability to transmit the virus and resist drugs, reminding us not to ignore the role of ticks as vectors in the prevention and control of multiple viruses. These findings indicate that H. longicornis have not independently acquired adaptive mutations that increase their adaptive capacity to environment and are not at risk of evolving into ‘super-ticks’” in line 412-423.

Round 2

Reviewer 1 Report

Comments and Suggestions for Authors

I have produced some conclusions for your idea. Try to rewrite them in the final version "The discovery of the broader distribution of H. longicornis throughout China and its relationship with to severe fever and thrombocytopenia syndrome virus (SFTSV) highlighted its potential in disease transmission. The biological characteristics and phylogenetic analysis revealed significant differences between the parthenogenetic and bisexual populations, suggesting a lack of gene flow and possible evolutionary advantages of parthenogenetic reproduction. The association of H. longicornis with blood parasites such as Theileria oriental and Rickettsia rickettsii, highlighted its role as a vector in disease transmission. It was critical to understand the mechanisms behind tick parthenogenesis, the evolutionary dynamics of tick populations, and the complex interactions between vectors, pathogens, and hosts". Some of text mistakes for examples species names in italics needed to be corrected. All figures needed to be replaced with higher resolution. 

Comments on the Quality of English Language

Moderate english corrections needed. 

Author Response

Reviewer 1:

Comments and Suggestions for Authors

I have produced some conclusions for your idea. Try to rewrite them in the final version "The discovery of the broader distribution of H. longicornis throughout China and its relationship with to severe fever and thrombocytopenia syndrome virus (SFTSV) highlighted its potential in disease transmission. The biological characteristics and phylogenetic analysis revealed significant differences between the parthenogenetic and bisexual populations, suggesting a lack of gene flow and possible evolutionary advantages of parthenogenetic reproduction. The association of H. longicornis with blood parasites such as Theileria oriental and Rickettsia rickettsii, highlighted its role as a vector in disease transmission. It was critical to understand the mechanisms behind tick parthenogenesis, the evolutionary dynamics of tick populations, and the complex interactions between vectors, pathogens, and hosts". Some of text mistakes for examples species names in italics needed to be corrected.

Answer: Thank you for the rewrite. We have replaced the main part of the conclusion with what you advised.

All figures needed to be replaced with higher resolution.

Answer: Thank you for pointing this out. We have improved the clarity of all figures.

Comments on the Quality of English Language

Moderate english corrections needed.

Reviewer 2 Report

Comments and Suggestions for Authors

Authors have correctly addressed my issues point by point, but some revisions are still needed. For example, at Line 76:

"Until now, there is no study which has systematically investigating the biological parameters of both parthenogenetic and bisexual populations in parallel, which holds great significance" please, check the use of English in the following sentence, such as "has systematically investigating".

Line 79 "we evaluated" would be better without personal forms.

Figure S1 has a legend that is out of focus; it needs to be improved.

Comments on the Quality of English Language

A quite good use of English, but some revisions are needed, as in the text highlighted in my comments.

Author Response

Reviewer 2:

Comments and Suggestions for Authors

Authors have correctly addressed my issues point by point, but some revisions are still needed. For example, at Line 76:

"Until now, there is no study which has systematically investigating the biological parameters of both parthenogenetic and bisexual populations in parallel, which holds great significance" please, check the use of English in the following sentence, such as "has systematically investigating".

Answer: Thank you for pointing this out. We have rephrased the word ‘investigating’ to ‘investigated’.

Line 79 "we evaluated" would be better without personal forms.

Answer: Thank you for pointing this out. We have rephrased the sentence ‘Here, we evaluated the differences between the parthenogenetic and bisexual populations of H. longicornis by comparing their distribution patterns, blood-meal times, biological and molecular characteristics’ to ‘Therefore, in this study, the differences between the parthenogenetic and bisexual populations of H. longicornis by comparing their distribution patterns, blood-meal times, biological and molecular characteristics were evaluated’ in line 78-81.

Figure S1 has a legend that is out of focus; it needs to be improved.

Answer: Thank you for pointing this out. We have revised the legend of Figure S1.

Comments on the Quality of English Language

A quite good use of English, but some revisions are needed, as in the text highlighted in my comments.
